# An Empirical Exploration of Open-Set Recognition via Lightweight Statistical Pipelines

## Abstract

Machine-learned safety-critical systems need to be self-aware and reliably *know their unknowns* in the open-world. This is often explored through the lens of anomaly/outlier detection or out-of-distribution modeling. One popular formulation is that of open-set classification, where an image classifier trained for 1-of-$K$ classes should also recognize images belonging to a $(K + 1)^{th}$ "other" class, *not* present in the training set. Recent work has shown that, somewhat surprisingly, most if not all existing open-world methods do not work well on high-dimensional open-world images (Shafaei et al., 2019). In this paper, we carry out an empirical exploration of open-set classification, and find that combining classic statistical methods with carefully computed features can dramatically outperform prior work. We extract features from off-the-shelf (OTS) state-of-the-art networks for the underlying $K$-way closed-world task. We leverage insights from the retrieval community for computing feature descriptors that are low-dimensional (via pooling and PCA) and normalized (via L2-normalization), enabling the modeling of training data densities via classic statistical tools such as kmeans and Gaussian Mixture Models (GMMs). Finally, we (re)introduce the task of open-set semantic segmentation, which requires classifying individual pixels into one of $K$ known classes or an "other" class. In this setting, our feature-based statistical models noticeably outperform prior open-world methods.

## 1 Introduction

Embodied perception and autonomy require systems to be self-aware and reliably *know their unknowns*. This requirement is often formulated as the *open set recognition* problem (Scheirer et al., 2012), meaning that the system, e.g., a $K$-way classification model, should recognize anomalous examples that do not belong to one of $K$ closed-world classes. This is a significant challenge for machine-learned systems that notoriously over-generalize to *anomalies* and *unknowns* on which they should instead raise a warning flag (Amodei et al., 2016).

**Open-world benchmarks:** Curating open-world benchmarks is hard (Liu et al., 2019). One common strategy re-purposes existing classification datasets into closed vs open examples – e.g., declaring MNIST digits 0-5 as closed and 6-9 as open (Neal et al., 2018; Oza & Patel, 2019; Geng et al., 2020). In contrast, anomaly/out-of-distribution (OOD) benchmarks usually generate anomalous samples by adding examples from different datasets - e.g., declaring CIFAR as anomalous for MNIST (Ge et al., 2017; Oza & Patel, 2019; Liu et al., 2019). Most open-world protocols assume open-world data is not available during training (Liang et al., 2018; Oza & Patel, 2019). Interestingly, Dhamija et al. (2018); Hendrycks et al. (2019b) find that, if some open examples are available during training, one can learn simple open-vs-closed binary classifiers that are remarkably effective. However, Shafaei et al. (2019) comprehensively compare various well-known open-world methods through rigorous experiments, and empirically show that *none* of the compared methods generalize to high-dimensional open-world images. Intuitively, classifiers can easily overfit to the available set of open-world images, which won't likely exhaustively span the open world outside the $K$ classes of interest.

In this paper, we carry out a rigorous empirical exploration of open-set recognition of high-dimensionial images. We explore simple statistical models such as Nearest Class Means (NCMs), kmeans and Gaussian Mixture Models (GMMs). Our hypothesis is that such classic statistical methods can reliably model the closed-world distribution (through the closed-world training data),

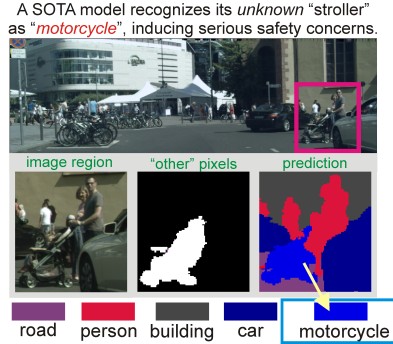 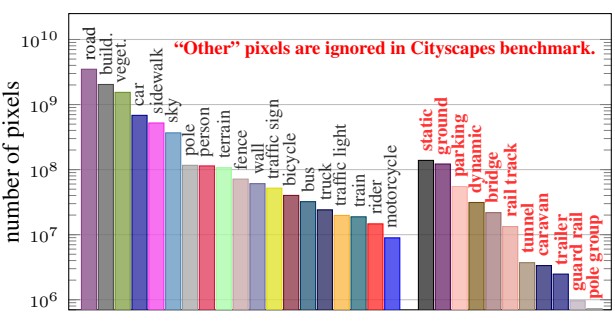

Figure 1: We motivate open-set recognition with safety concerns in autonomous systems. **Left**: State-of-the-art semantic segmentation networks (Wang et al., 2019) do not model "strollers", which are outside the $K$ closed-set categories in Cityscapes benchmark (Cordts et al., 2016). Here, the network misclassifies the "stroller" as a "motorcycle", which can be a critical mistake when fed into an autonomy stack because the two objects exhibit different behaviours (and so require different plans for obstacle avoidance). **Right**: While classic semantic segmentation benchmarks explicitly evaluate background pixels outside the set of $K$ classes (Everingham et al., 2015), contemporary benchmarks such as Cityscapes ignore such pixels during evaluation. As a result, most segmentation networks also ignore such pixels during training. Perhaps surprisingly, such ignored pixels include vulnerable objects like wheelchairs and strollers (see left). We repurpose these ignored pixels as open-set examples that are from the $(K+1)^{th}$ "other" class, allowing for a large-scale exploration of open-set recognition via semantic segmentation.

and help avoid overfitting (an issue in open-vs-closed classifiers). Traditionally, such simple models have been used to address the open-world (Chandola et al., 2009; Geng et al., 2020), but are largely neglected in the recent literature. We revisit these simple methods, and find them quite effective once crucial techniques are considered, as summarized by contributions below.

**Contribution 1:** We build classic statistical models on top of off-the-shelf (OTS) features computed by the underlying $K$-way classification network. We find it crucial to use OTS features that have been pre-trained and post-processed appropriately (discussed further below). Armed with such features, we find classic statistical models such as kmeans and GMMs (Murphy, 2012) can outperform prior work. We describe two core technical insights below.

**Insight-1 Pre-training** networks (e.g., on ImageNet (Deng et al., 2009)) is a common practice for traditional closed-world tasks. However, to the best of our knowledge, open-world methods do not sufficiently exploit pre-training (Oza & Patel, 2019). Hendrycks et al. (2019a) report that pre-training improves anomaly detection using softmax confidence thresholding (Hendrycks & Gimpel, 2017). We find pretraining to be a crucial factor in learning better representations that support more sophisticated open-world reasoning. Intuitively, pre-trained networks expose themselves to diverse data that may look similar to open-world examples encountered at test-time. We operationalize this intuition by building statistical models on top of *existing* discriminative networks, which tend to make use of pre-training by design. We demonstrate this significantly outperforms features trained from scratch, as most prior open-set work does.

**Insight-2 Low-dimensional normalized features.** While some existing open-world methods also exploit OTS features (Lee et al., 2018), we find it crucial to make use of insufficiently well-known best practices for feature extraction. Specifically, to reduce dimensionality, we pool spatially (Gong et al., 2014) and use principle component analysis (PCA) (Turk & Pentland, 1991). Then, to ensure features are invariant to scalings, we adopt L2 normalization (Gong et al., 2014; Gordo et al., 2017). While these are somewhat standard practices for deep feature extraction in areas such as retrieval, their combination is not well explored in the open-set literature (Bendale & Boult, 2016; Grathwohl et al., 2019). Given a particular OTS $K$-way classification network, we determine the "right" feature processing through validation. In particular, we find that L2-normalization greatly boosts open-world recognition performance; spatial pooling and PCA altogether reduce feature dimension by three orders of magnitude without degrading performance, resulting in a lightweight pipeline.

**Contribution 2:** We re(introduce) the problem of open-set semantic segmentation. Interestingly, classic benchmarks explicitly evaluate background pixels outside the set of $K$ classes of interest (Everingham et al., 2015). However, contemporary benchmarks such as Cityscapes (Cordts et al., 2016)

Figure 2: Flowchart for extracting off-the-shelf (OTS) features used for open-set recognition. We determine the appropriate feature processing steps on validation set, including spatial pooling (sp) and L2 normalization (L2). **Left**: for open-set image recognition, we extract OTS features at the last convolution layer of a $K$-way classification network. **Right**: for open-set semantic segmentation, we extract OTS features from the "pyramid head" module, which has sufficiently captured multi-scale information. We do not adopt spatial pooling and instead use the per-pixel features to represent pixels. Note that, different from our practice, many other methods like OpenMax (Bendale & Boult, 2016) and generative methods (Grathwohl et al., 2019) work on logit features, which are too invariant to be effective for open-set recognition (cf. Figure 3 and Table 2).

ignore such pixels during evaluation. As a result, most contemporary segmentation networks also ignore such pixels during training. Perhaps surprisingly, such ignored pixels include vulnerable objects like strollers and wheelchairs. Misclassifying such objects may have serious implications for real-world autonomous systems (see Figure 1). Instead of ignoring these pixels, we use them to explore open-world recognition by repurposing them as open-world examples. Interestingly, this setup naturally allows for open-pixels in the train-set, a protocol advocated by (Dhamija et al., 2018; Hendrycks et al., 2019b). We benchmark various open-world methods on this setup, and show that our suggested simple statistical models still outperform typical open-world methods. Similar to past work, we also find that simple open-vs-closed binary classifiers serve as strong baselines, provided one has enough training examples of open pixels that span the open-world.

## 2 RELATED WORK

**Open-set recognition**. There are multiple lines of work addressing the open-world problems in the context of $K$-way classification, such as anomaly/out-of-distribution detection (Chandola et al., 2009; Zong et al., 2018; Hendrycks et al., 2019b), novelty/outlier detection (Pidhorskyi et al., 2018). Defined on $K$-way classification, these problems can be crisply formulated as open-set recognition (Scheirer et al., 2012; Bendale & Boult, 2016; Lee et al., 2018; Geng et al., 2020). Given a testing example, these methods compute the likelihood that it belongs to the open-world via post-hoc functions like density estimation (Zong et al., 2018), uncertainty modeling (Gal & Ghahramani, 2016; Liang et al., 2018; Kendall & Gal, 2017) and reconstruction error of the testing example (Pidhorskyi et al., 2018; Dehaene et al., 2020). Different from the above sophisticated methods, we train simple statistical models (e.g., GMM) which can work much better by following our proposed pipeline.

**Feature extraction.** Off-the-shelf (OTS) features can be extracted from the discriminative network and act as powerful embeddings (Donahue et al., 2014). Using OTS features for open-set recognition has been explored in prior work (Oza & Patel, 2019; Grathwohl et al., 2019; Lee et al., 2018). OTS features can be logits, softmax and other intermediate feature activations computed by the discriminative network. Early open-set methods modify the softmax (Hendrycks & Gimpel, 2017; Bendale & Boult, 2016). Grathwohl et al. (2019) learn an energy-based model over the logit features for anomaly detection. Oza & Patel (2019) reconstruct input images from penultimate-layer features and use the reconstruction error as the open-set likelihood. Most related to our work is Lee et al. (2018), who build Gaussian models over OTS features for anomaly detection, but relies on input image perturbation for better open-set classification performance. In contrast, we study even simpler statistical models such as kmeans and GMM, and show that proper feature processing (via L2-normalization and PCA) greatly boosts the efficacy and efficiency of open-set recognition.

## 3 OPEN-SET RECOGNITION VIA LIGHTWEIGHT STATISTICAL PIPELINES

In this section, we discuss various design choices in our pipeline, including (1) training schemes for the underlying closed-world task, (2) methods for extracting and repurposing closed-world feature descriptors for open-world recognition, and (3) the statistical density estimation models built on such

extracted features. We conclude with (4) an analysis of the additional compute required for self-aware processing (via the addition of an open-world "head" on top of the closed-world network), pointing out that minimal additional processing is needed.

**1. Network training strategies.** Virtually all state-of-the-art deep classifiers make use of large-scale pre-training, e.g., on ImageNet (Deng et al., 2009), which seems to consistently improve towards the state-of-the-art performance on the closed-world data (Sun et al., 2017; Mahajan et al., 2018). However, many, if not all, open-world methods trains the discriminant network purely on the closed-world data *without* pre-training (Oza & Patel, 2019; Hendrycks & Gimpel, 2017). We argue that a pre-trained network also serves as an abstraction of the (pseudo) open world. Intuitively, such a pre-trained model has already seen diverse data that may look similar to the open-world examples that will be encountered at test-time, particularly if ImageNet does not look similar to the (closed) training set for the task of interest. Recently, Hendrycks et al. (2019a) show that pre-training improves open-world robustness with a simplistic method that thresholds softmax confidence (Hendrycks & Gimpel, 2017). Our diagnostic study shows that our explored statistical models, as well as prior methods, do perform much better when built on a pre-trained network than a network trained from scratch!

**2. Feature extraction.** OTS features generated at different layers of the trained discriminative model can be repurposed for open-set recognition (Lee et al., 2018). Most methods leverage softmax (Hendrycks & Gimpel, 2017) and logits (Bendale & Boult, 2016; Grathwohl et al., 2019) which can be thought of as features extracted at top layers. Similar to (Lee et al., 2018), we find it crucial to analyze features from intermediate layers for open-set recognition, for which logits and softmax may be too invariant to be effective for open-set recognition (see Figure 3). One immediate challenge to extract features from an intermediate layer is their high dimensionality, e.g., of size 512x7x7 from ResNet18 (He et al., 2016). To reduce feature dimension, we simply (max or average) pool the feature activations spatially into a 512-dim feature vectors (Yang & Ramanan, 2015). We further use PCA, which can reduce dimension by $10\times$ (from 512-dim to 50-dim) without sacrificing performance. We find this dimensionality particularly important for learning second-order covariance statistics as in GMMs, described below. Finally, following (Gong et al., 2014; Gordo et al., 2017), we find it crucial to L2-normalize extracted features (see Figure 2).

**3. Statistical models.** Given the above extracted features, we can learn various generative statistical models to capture the confidence/probability that a test example belongs to the closed-world distribution. We explore simple parametric models such as Nearest Class Means (NCMs) (Mensink et al., 2013) and class-conditional Gaussian models (Lee et al., 2018; Grathwohl et al., 2019), as well as non-parametric models such has nearest neighbors (NN) (Boiman et al., 2008; Júnior et al., 2017). We finally explore an intermediate regime of mixture models, including (class-conditional) GMMs and kmeans (Chandola et al., 2009;?; Cao et al., 2016; Geng et al., 2020). Our models label a test example as open-world when the inverse probability (e.g., of the most-likely class-conditional GMMs) or distance (e.g., to the closest class centroid) is above a threshold. One benefit of such simple statistical models is that they are interpretable and relatively easier to diagnose failures. For example, one failure mode is an open-world sample being misclassified as a closed-world class. This happens when open-world data lie close to a class-centroid or Gaussian component mean (see Figure 3-left). Note that a single statistical model may have several hyperparameters – GMMs can have multiple Gaussian components and different structures of second-order covariance, e.g., either a single scalar, a vector or a full-rank general covariance per component, as denoted by *"spherical"*, *"diag"* and *"full"*, respectively. We make use of a validation set to determine the hyperparameters (as well as feature processing steps listed above).

**4. Lightweight Pipeline.** We re-iterate that the above feature extraction and statistical models result in a lightweight pipeline for open-set recognition. We now analyze the number of additional parameters in our pipeline. Naively learning a GMM over features from the last convolutional layer result in massive second-order statistics, on the order of $(512 \times 7 \times 7)^2$ for a 512x7x7 Res18 feature map. We find that spatial pooling and PCA can reduce dimensionality to $50$, which requires only $50^2$ covariance parameters (a reduction of $10^5$). We find linear dimensionality reduction more effective than sparse covariance matrices (e.g., assuming diagonal structure). The appendix includes additional experiments. Given a class-conditional five-component GMM (the largest found to be effective through cross validation), this requires 128KB storage per class, or 594KB for all 19 classes in Cityscapes. This is less than 0.1% of the compute of the underlying closed-world network

(e.g., HRNet at 250 MB), making it a quite practical addition that enables self-aware processing on real-time autonomy stacks.

## 4 EXPERIMENT

We extensively validate our proposed lightweight statistical pipeline under standard open-set recognition benchmarks, typically focused on image classification. We also consider open-set semantic segmentation, revisiting classic formulations of semantic segmentation that make use of a background label (Everingham et al., 2015). We start by introducing implementation details, evaluation metrics and baselines. We then present comprehensive evaluations on each setup.

**Implementation.** As discussed earlier, open-world recognition is often explored through the lens of open-set classification. To ensure our approaches retain high-accuracy on the original closed-world tasks, we build statistical models on top of off-the-shelf (OTS) state-of-the-art networks. For open-set image classification, we fine-tune an ImageNet-pretrained ResNet network (Res18/50 in our experiments) (He et al., 2016) exclusively on the closed-train-set using cross-entropy loss. For open-set semantic segmentation we use HRNet (Wang et al., 2019), a highly-ranked model on the Cityscapes leaderboard (Cordts et al., 2016). We extract features at the penultimate layer of each discriminative network (other layers also apply but we do not explore them in this work). We conduct experiments with PyTorch (Paszke et al., 2017) on a single Titan X GPU.

**Evaluation Metric**. Following past work (Hendrycks & Gimpel, 2017; Lee et al., 2018), we evaluate binary detection of open-world examples using the area under the receiver operating characteristic curve (AUROC) (Davis & Goadrich, 2006). AUROC is a calibration-free and threshold-less metric, simplifying comparisons between methods. For open-set semantic segmentation, we also use AUROC to evaluate the performance of recognizing "background" pixels as open-world examples. This is different from traditional practice in segmentation benchmarks (Everingham et al., 2015) which treat such "background" pixels as just another class.

**Baselines.** Our statistical pipeline supports various statistical models. We study the simple models proposed in Section 3, including NN, kmeans, NCM, and GMMs. All models, including baselines to which we compare, are based on the same underlying classification network. Hyperparameters for all models (e.g., number of mixtures) are tuned on a validation set[1].

- **Classifiers.** Hendrycks et al. (2019b) learn a binary open-vs-closed classifier ($CLS^2$) for anomaly detection. Following classic work in semantic segmentation (Everingham et al., 2015), we also evaluate a $(K+1)$-way classifier ($CLS^{(K+1)}$). We use the softmax score corresponding to the $(K+1)^{th}$ "other" class as the open-set likelihood. Both methods require open-set examples during training.

- **Likelihoods.** Many probabilistic models measure open-set likelihood on OTS features, including Max Softmax Probability (MSP) (Hendrycks & Gimpel, 2017) and Entropy (Steinhardt & Liang, 2016) (derived from softmax probabilities). OpenMax (Bendale & Boult, 2016) fits logits to Weibull distributions (Scheirer et al., 2011) that recalibrate softmax outputs for open-set recognition. C2AE (Oza & Patel, 2019) learns an additional $K$-way classifier on OTS features based on reconstruction errors, which are then used as open-set likelihood function. GDM (Lee et al., 2018) learns a Gaussian Discriminant Model on OTS features and designs open-set likelihood based on Mahalanobis distance. CROSR (Yoshihashi et al., 2019) trains a reconstruction-based model that jointly performs closed-set $K$-way classification and open-set recognition. G-Open (Ge et al., 2017) and OSRCI (Neal et al., 2018) turn to Generative Adversarial Networks (GANs) to generate fake images that augment the closed-set training set, and train a discriminative model for open-set recognition. CGDL (Sun et al., 2020) learns class-conditional Gaussian model and relies on reconstruction error for open-set recognition. The last three methods (CROSR, G-Open and CGDL) train ground-up models in contrast to our statistical models that operate on OTS features of an *already-trained* $K$-way classification network. As we focus on an empirical exploration rather than achieving the state-of-the-art, we refer readers to more recent approaches for the state-of-the-art by training ground-up models with sophisticated techniques (Zhang et al., 2020; Chen et al., 2020).

---

[1]We use open-source code when available. We implemented C2AE and its authors validated our code through personal communication.

Table 1: **Single-dataset open-set recognition (Setup-I) AUROC**↑. Error bars are shown in gray rows . Bolded numbers mark the top-5 ranked methods on each dataset. Because GDM does not L2-normalize features, we add a variant that does (denoted GDM$_{L2}$). L2-normalization clearly improves performance, particularly on CIFAR. Interestingly, GDM$_{L2}$ underperforms NCM, implying that a full-covariance Gaussian (used by GDM) overfits compared to an identity covariance. GMM makes use of low-dimensional covariances (learned via PCA) and strikes a balance between flexibility and generalization, achieving comparable performance to many sophisticated approaches.

| | MSP | Entropy | OpenMax | G-Open | OSRCI | CROSR | C2AE | CGDL | MSP$_c$ | MCdrop | GDM | GDM$_{L2}$ | NN | NCM | kmeans | GMM |
|---|---|---|---|---|---|---|---|---|---|---|---|---|---|---|---|---|
| *MNIST* | .977 | .988 | .981 | .984 | .988 | **.998** | .989 | **.994** | .985 | .981 | .984 | .984 | .989 | **.991** | **.991** | **.993** |
| | .002 | .002 | .002 | .005 | .004 | .004 | .002 | .002 | .002 | .002 | .002 | .002 | .002 | .002 | .002 | .002 |
| *SVHN* | .886 | .895 | .894 | .896 | **.910** | **.955** | **.922** | **.935** | .891 | .888 | .862 | .868 | .866 | .901 | .906 | **.914** |
| | .006 | .006 | .013 | .017 | .006 | .004 | .009 | .003 | .006 | .005 | .005 | .003 | .003 | .003 | .004 | .003 |
| *CIFAR* | .757 | .788 | **.811** | .675 | .699 | — | **.895** | **.903** | .808 | .801 | .686 | .744 | .752 | .798 | **.811** | **.817** |
| | .032 | .030 | .032 | .035 | .029 | — | .008 | .009 | .015 | .011 | .009 | .008 | .007 | .007 | .008 | .008 |

- **Bayesian Networks.** Bayesian neural networks compute uncertainty estimates via Monte Carlo estimates (MCdrop) (Gal & Ghahramani, 2016; Loquercio et al., 2020) and calibrated Max Softmax Probability (MSP$_c$) (Liang et al., 2018), which can also be used as open-set likelihoods. We implement MCdrop via 500 samples.

## 4.1 SETUP-I: SINGLE-DATASET OPEN-SET RECOGNITION

**Setup.** We begin by following the standard benchmark protocol used in most prior work; split a single dataset into open and closed sets w.r.t class labels (e.g., define MNIST digits 0-5 as the closed-set, and digits 6-9 as the open-set). This is a common practice in open-set recognition (Neal et al., 2018; Oza & Patel, 2019). Notably, methods do not have access to open-set examples during training.

**Datasets.** MNIST / CIFAR / SVHN are popular datasets used in the open-set recognition literature (Neal et al., 2018; Hendrycks & Gimpel, 2017). All three datasets contain ten classes with balanced numbers of images per class. Standard protocol randomly splits six (four) classes of train/validation-sets into closed (open) train/validation-sets, respectively. We repeat five times and report average AUROC for each method. Through cross-validation, we find reliable OTS features can be computed by average-pooling features from the last convolutional layer down to 512-dim, projecting down to 50-dim via PCA, and L2-normalizing.

**Results.** Table 1 shows that, perhaps surprisingly, simple statistical models (like kmeans and GMMs) defined on such normalized features already performs on par to many prior methods/ Because GDM (Lee et al., 2018) does not L2-normalize features, we evaluate a variant that does (GDM$_{L2}$). The improved performance demonstrates the importance of feature normalization, which although is well known in the image retrieval community, is not widely used in open-set recognition. We hereby focus on statistical models trained on normalized features, providing raw vs. normalized comparisons in the appendix.

## 4.2 SETUP-II: CROSS-DATASET OPEN-SET RECOGNITION

**Setup.** In these experiments, we use the *cross-dataset* protocol advocated by (Shafaei et al., 2019), where *some* outlier examples are sampled from a different dataset for training/validation. (e.g., train on TinyImageNet-closed as closed-set, validate on MNIST-open as outlier set, and test on CIFAR-open as open-set). Conclusions drawn under this setup may generalize better due to less dataset bias in the experimental protocol (Torralba & Efros, 2011).

**Datasets**. We use TinyImageNet as the closed-world dataset (for $K$-way classification), which has 200 classes of 64x64 images, split into 500/50/50 images as the train/val/test sets. Following (Shafaei et al., 2019), we construct open val/test sets using cross-dataset images (Torralba & Efros, 2011), including MNIST, SVHN, CIFAR and Cityscapes. For example, we use an outlier dataset (e.g., MNIST train-set) to tune/train an open-world method, and test on another dataset as the open-set (e.g., CIFAR test-set). We use bilinear interpolation to resize all images into 64x64 to match TinyImageNet image resolution. Through cross validation, we find reliable OTS features can be computed by average-pooling features from the last convolutional layer down to 2048-dim, projecting down to 200-dim via PCA, and L2-normalizing.

**Results** for Table 2 are summarized below:

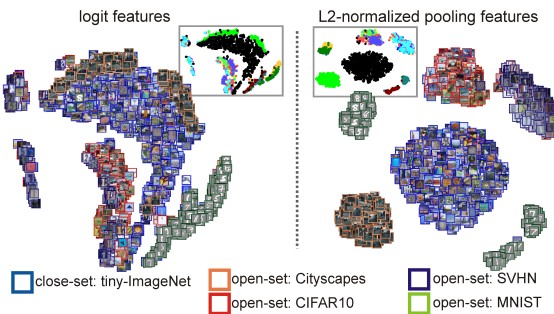

logit features    L2-normalized pooling features

□ close-set: tiny-ImageNet    □ open-set: Cityscapes    □ open-set: SVHN
□ open-set: CIFAR10    □ open-set: MNIST

Figure 3: tSNE plots (Maaten & Hinton, 2008) of open-vs-closed data, as encoded by different features from a Res50 model (trained with pre-training in the closed world, cf. Table 2). Points are colored w.r.t closed-world class labels. **Left**: Logit features mix open and closed data, suggesting that methods based on them (Entropy, SoftMax and OpenMax) may struggle in open-set classification. **Right**: Convolutional features better separate open vs closed data (cf. Figure 2).

Table 2: **Cross-dataset open-set image recognition (Setup-II) AUROC↑**. In this setup, we train on Tiny-ImageNet, validate using outlier images from a second dataset, and test using open-set images from a third dataset. For each open-set dataset, we compute the average AUROC over all results when using different outlier datasets. We study two Res50 models either trained from scratch (pink row), or fine-tuned from an ImageNet-pretrained model (blue row). Clearly, simple statistical models can handily outperform much prior work. Pre-training boosts open-set recognition performance for *all* methods (see last row pair). Binary classifiers $CLS^2$ do not generalize well, presumably due to overfitting. Somewhat surprisingly, OpenMax works quite poorly. We conjecture that the regularized logit features on which it is based may too invariant to be effective for cross-dataset open-set recognition. Table 4 and 5 supplement this table with more details.

| open-test | MSP | Entropy | OpenMax | $MSP_c$ | MCdrop | C2AE | GDM | $GDM_{L2}$ | NN | NCM | kmeans | GMM | $CLS^2$ | $CLS^{(K+1)}$ |
|---|---|---|---|---|---|---|---|---|---|---|---|---|---|---|
| MNIST | .709 | .712 | .144 | .773 | .657 | .811 | .454 | .799 | **.966** | .961 | .939 | .940 | .963 | .939 |
|  | .775 | .789 | .453 | .832 | .801 | .796 | .723 | .957 | .901 | **.979** | .963 | .964 | **.986** | .944 |
| SVHN | .752 | .768 | .314 | .803 | .833 | .723 | .841 | .991 | .993 | **.994** | .982 | .984 | .754 | .907 |
|  | .770 | .787 | .123 | .863 | .783 | .780 | .820 | **.999** | .994 | .995 | .993 | .990 | .701 | .948 |
| CIFAR | .694 | .703 | .338 | .750 | .741 | .719 | .712 | .886 | .852 | .963 | .937 | **.968** | .739 | .867 |
|  | .725 | .732 | .471 | .791 | .809 | .763 | .838 | .961 | .927 | **.975** | .948 | .961 | .754 | .880 |
| Citysc. | .739 | .753 | .604 | .862 | .877 | .753 | .725 | .650 | .559 | .839 | **.903** | .885 | .601 | **.919** |
|  | .751 | .762 | .543 | .851 | .868 | .784 | .651 | .513 | .715 | .833 | **.886** | .867 | .646 | **.971** |
| *average* | .723 | .734 | .350 | .797 | .777 | .752 | .683 | .832 | .843 | .939 | **.940** | .938 | .764 | .908 |
|  | .755 | .768 | .397 | .834 | .815 | .781 | .758 | .857 | .884 | .946 | **.948** | .945 | .772 | .936 |

- Simple statistical models (e.g., NCM and kmeans) can outperform prior open-set methods (e.g., C2AE and GDM). We find that L2-normalization greatly contributes to the success of these simple statistical methods (cf. details in the appendix). Both the metric learning and image retrieval (Mensink et al., 2012; Musgrave et al., 2020) literature have shown the importance of L2-normalization. Informally, open-set recognition queries the testing example and measures how close it is to any of the closed-world training examples (Musgrave et al., 2020).

- Interestingly, kmeans performs slightly better than GMMs. Considering that the former can be seen as a special case of GMMs that have an identity covariance, we conjecture that learning other types of covariance (e.g., a full-rank covariance matrix) does not help when the underlying $K$-way network has already provided compact feature representations.

- From last row pair, we can see pre-training notably improves all the methods. $GDM_{L2}$ outperforms the original GDM which operates on raw features (without L2-normalization). This further confirms the importance of L2-normalization in feature extraction for open-set recognition.

- Perhaps surprisingly, OpenMax does not work well in this setup (though we have spent considerable effort tuning it). This is consistent with the results in (Dhamija et al., 2018; Shafaei et al., 2019), and we conjecture the reason is that OpenMax cannot effectively recognize *cross-dataset anomalous inputs* using logit features because they are too invariant to be useful for open-set recognition (Figure 3). Similar lackluster results hold for other methods that operate on logit features (Entropy and MSP).

## 4.3 SETUP-III: OPEN-SET SEMANTIC SEGMENTATION

**Setup.** In these experiments, we (re)introduce the task of open-set segmentation by repurposing "background" pixels in contemporary segmentation benchmarks (Cityscapes) as open-world pixels. As elaborated before, such pixels are either traditionally treated as just another class for segmentation evaluation (Everingham et al., 2015) or ignored completely. Instead, we evaluate them using open-world metrics such as AUROC. We will show our statistical methods also outperform other typical open-world methods. As this setup has natural access to open-world pixels during training, we explore the training of simple open-vs-closed classifiers.

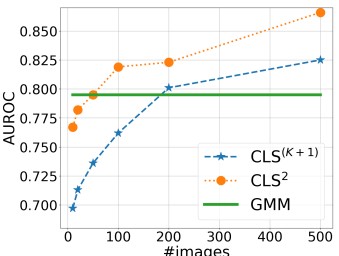

Figure 4: Two random images from Cityscapes-val, visualized with ground-truth and the predicted semantic segmentation maps by the HRNet. We visualize open-world pixels in the ground-truth (white regions), as well as predicted open-world pixels for a standard baseline (MSP) and our method (GMM). MSP tends to predict segment boundaries as open-pixels, while our GMM tends to find open-set objects (street-shop and rollator as pointed by the red arrows).

Figure 5: Performance of CLS versus amount of (open) training data. We train CLS on the OTS features of the state-of-the-art HRNet, which has already exploited all closed-world pixels in the train-set. The binary open-vs-closed classifier $CLS^2$ outperforms $CLS^{(K+1)}$, presumably due to that the former being trained on balanced batches. With fewer than 50 images, GMMs outperform such discriminative models. But with enough open training examples, simple binary classification performs remarkably well. However, because GMMs do not require any open training examples, it cannot overfit to them and so may generalize better to the open-world.

**Datasets**. Cityscapes (Cordts et al., 2016) provides per-pixel annotations for urban scene images (1024x2048-resolution) for autonomous driving research. We construct our train- and val-sets from its 2,975 training images, in which we use the last 10 images as val-set and the rest as train-set. We use its official 500 validation images as our test-set. The "background" pixels (shown in white of ground-truth visual in Figure 4) are the open-world examples in this setup. Through validation, we find reliable OTS features can be computed by projecting features from the last convolutional layer from 720 down to 100-dim via PCA, and L2-normalizing.

**Results.** For our statistical models (as well as GDM), we randomly sample 5000 closed-world pixel features from each class, as it is prohibitively space-consuming to use all the pixel features from the Cityscapes train-set. We show quantitative comparison in Table 3 and list salient conclusions below.

- Clearly, our simple statistical models (e.g., NN and GMM) perform significantly better than the classic open-world methods (e.g., MSP and OpenMax). However, when training on large amounts of open-pixels, CLS methods achieve significantly better performance. This clearly shows the benefit of training on open-world pixels (Hendrycks et al., 2019b). We do note that GMMs do not need *any* open pixels during learning, and so may generalize better to novel open-world scenarios not encountered in the training set (Figure 5).

- GDM performs poorly, probably due to arbitrary scales of the raw features that are too uninformative to be used for open-set pixel recognition. We note that other statistical methods all struggle with raw pooled features (cf. appendix). However, once we L2-normalize the pixel features to be scale-invariant, these statistical methods perform significantly better (as reported in this table).

- Figure 4 shows qualitative results. MSP predicts segment boundaries as open-pixels. This makes sense as the MSP mostly returns aleatoric uncertainties corresponding to ambiguous pixel sensor measurements around object boundaries (Kendall & Gal, 2017). In contrast, GMM reports open-pixels on truly novel objects, such as the `street-shop` and `rollator`, both of which are ignored by the semantic segmentation network during training HRNet (Wang et al., 2019). These regions appear to be caused by epistemic uncertainty arising from the lack of training data (Kendall & Gal, 2017).

- Figure 6 plots AUROC performance vs model size for various statistical models. Notably, NN consumes the most memory, even more than the underlying networks. GMMs perform the best and are quite lightweight, only consuming 0.6MB when built on the HRNet model (250MB).

Table 3: **Open-set semantic segmentation (Setup-III) AUROC↑**. Simple statistical methods (GMMs) outperform prior methods, with the notable exception of discriminative classifiers (CLS$^2$ and CLS$^{(K+1)}$) that have access to open-set training examples. Figure 5 analyzes this further, demonstrating that GMMs can outperform such discriminative models when they have access to less open training examples, suggesting that GMMs may better generalize to never-before-seen open-world scenarios.

| MSP | Entropy | OpenMax | C2AE | MSP$_c$ | MCdrop | GDM | NN | NCM | kmeans | GMM | CLS$^2$ | CLS$^{(K+1)}$ |
|-----|---------|---------|------|---------|--------|-----|-----|------|--------|------|---------|---------------|
| .590 | .600 | .655 | .603 | .612 | .563 | .539 | .769 | .715 | .755 | **.795** | .897 | .867 |

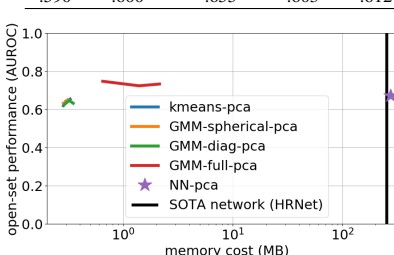

Figure 6: AUROC vs. memory cost (MB) for various statistical models for open-set semantic segmentation. NN stores ∼100k OTS features, which is larger than the underlying network (HRNet). We explore GMMs with various covariance structures (spherical, diagonal, full), feature dimensionality via PCA, and mixture components. We find the best AUROC-memory tradeoff on the validation set (shown here to be a single-mixture GMM with full-covariance and PCA), and find it generalizes well to held-out test (cf. Appendix).

## 5 CONCLUSION

We explore an empirical exploration of open-set recognition via lightweight statistical pipelines. We find simple statistical models quite effective if built on properly processed off-the-shelf features computed by the discriminative networks (originally trained for the closed-world tasks). Our pipelines endow $K$-way networks with the ability to be "self-aware", with negligible additional compute costs (0.1%). Finally, we (re)introduce the task of open-set semantic segmentation by repurposing background pixels as open-world examples, requiring classification of individual pixels into one of $K$ known/closed-world classes and an "other" open-world class.

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

APPENDIX OUTLINE

As elaborated in the main paper, we introduce a lightweight statistical pipeline for open-set recognition by repurposing off-the-shelf (OTS) features computed by a state-of-the-art recognition network. As our pipeline does not require (re)training the underlying network, it is guaranteed to replicate the state-of-the-art performance of the network on the (closed-world) task for which it was trained, but still allows the final recognition system to properly identify never-before-seen data from the open-world. In the appendix, we expand on our pipeline, including more experiments, analyses and visualizations. We outline the appendix below.

**Section A: Data statistics for open-set semantic segmentation.** We provide data details used for open-set semantic segmentation (Setup-III), motivated by safety concerns in autonomous stacks as shown in Figure 1 left.

**Section B: Detailed results by statistical models.** We provide detailed results on the open-set recognition including open-set image recognition (Setup-II), and open-set semantic segmentation (Setup-III). We detail the performance of the various statistical models studied in the main paper, including Nearest Neighbor (NN), Nearest Class Mean (NCMs), kmeans and Gaussian Mixture Models (GMMs).

**Section C: Reduced dimension via PCA.** We show that PCA can reduce dimensionality significantly (making our pipeline quite lightweight), while maintaining or even improving performance.

**Section D: Performance vs. memory/compute.** We rigorously evaluate the memory/compute costs of our various statistical pipelines, emphasizing solutions that are both accurate and lightweight.

**Section E: Visualization of Gaussian component means.** One benefit of our simple statistical models is their interpretability; we visualize Gaussian means through centroid images, and demonstrate that they correspond to canonical objects (e.g., those with standard poses and clean background).

**Section F: Open-Source demonstration.** We include code (via Jupyter Notebook) for open-set semantic segmentation, assuming one has access to precomputed features from HRNet (Wang et al., 2019).

## A    SETUP FOR OPEN-SET SEMANTIC SEGMENTATION

As we (re)introduce the task of open-set semantic segmentation for exploring open-set recognition, in which we re-purpose "backgroud" pixels of Cityscapes as open-world examples (that are from the $(K+1)^{th}$ "other" class). We hereby list the statistics of open and closed-world examples (pixels). Cityscapes training set has 2,975 images. We use the first 2,965 images for training, and hold out the last 10 as validation set for model selection. We use the 500 Cityscapes validation images as our test set. Here are the statistics for the full train/val/test sets.

- train-set for closed-pixels: 2,965 images providing 334M closed-set pixels.
- train-set for open-pixels: 2,965 images providing 44M open-set pixels.
- val-set for closed-pixels: 10 images providing 1M closed-set pixels.
- val-set for open-pixels: 10 images providing 0.2M open-set pixels.
- test-set for closed-pixels: 500 images providing 56M pixels.
- test-set for open-pixels: 500 images providing 8.3M pixels.

## B    DETAILED RESULTS BY STATISTICAL PIPELINE

In Figure 10 on the last page of this document, we provide detailed results of various statistical models for open-world tasks, including open-set image recognition and open-set semantic segmentation. In the main paper, we state that we tune and select statistical models (if it has hyper-parameters to tune) on the small validation set, and report on the test set with the selected (best-performing) model. Such hyper-parameters can be the number of means/components in kmeans and GMM models, and covariance types in GMM — "spherical", "diagonal" and "full" denote that the covariance matrix of each Gaussian component is controlled by a single scalar, a vector and full-rank matrix, respectively. It demonstrates that validation can reliably tune the statistical models whose performance can be translated to the test sets. Moreover, we also record the detailed results of whether using

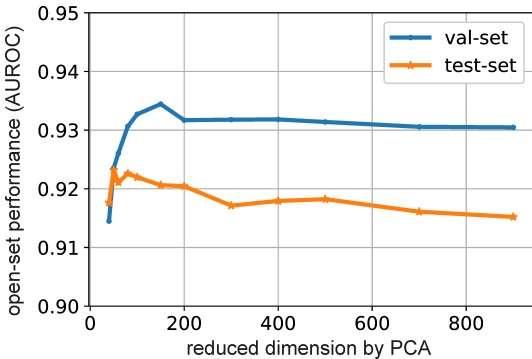

Figure 7: **Study of open-set performance versus feature dimension reduced by PCA**. We study this with the experiment of open-set image classification, where TinyImageNet/Cityscapes are the closed/open sets. We extract OTS features from the Res50pt network (as detailed in the paper). The spatially pooled (and L2-normalized) feature has 2048 dimension. To avoid randomness (as in some statistical models like GMM and k-means), we use NCM model which simply computes per-class mean feature and computes the distance of nearest center as the open-set likelihood. Surprisingly, using PCA to reduce feature dimension can improve the open-set performance!

L2-normalization on the features. We can see that L2-normalization greatly boosts open-world performance.

Table 4 and 5 list details of various methods on cross-dataset evaluation (Setup-II), supplementing Table 2. Please refer to the caption for details.

## C  PERFORMANCE VS. PCA REDUCED DIMENSION

As analyzed in the main paper, PCA is an important technique to make our pipeline lightweight by considerably reducing feature dimensions. We study how a statistical model performs under different reduced feature dimensions by PCA. We choose the simplistic NCM method which does not induce randomness (unlike kmeans and GMM which require random initialization for learning). We study this through open-set image recognition under Setup-II. To simplify the study, we choose (resized) Cityscapes images as open-set data, i.e., we use TinyImagenet/Cityscapes images as the closed/open set. As we use the network Res50 in the diagnostic study, the original dimension of the pooled features is 2048. In Fig. 7, we plot the performance (AUROC) of NCM as a function of reduced dimension by PCA. Perhaps surprisingly, PCA even improves the open-world performance while significantly reducing feature dimension (from 2048 to 100)!

## D  PERFORMANCE VS. MEMORY/COMPUTE

As seen previously, PCA reduces the feature dimension greatly and hence makes the statistical models quite lightweight. We now study how lightweight different statistical models can be by considering the open-world performance. We analyze the models learned for two tasks (open-world image classification and open-world semantic segmentation), where the OTS features have dimension 2048 (extracted from Res50) and 720 (extracted from HRNet), respectively. We use PCA to reduce the features dimensions to 200 and 100, respectively.

We focus on NN, kmeans and GMMs, all of which operate on the PCA reduced features (with L2-normalization). NN is the straightforward baseline that memorizes all training examples to recognize open-set examples. For GMM, we study it by specifying three types of covariance – "spherical", "diag" and "full" meaning the covariance matrix of each Gaussian component is controlled by a single scalar, a vector or a full-rank (symmetric) matrix, respectively. For open-set semantic segmentation, it is prohibitively space-consuming to memorize all the pixel features of the whole train set. So we randomly sample 5000 pixels from each of the 19 classes defined by Cityscapes (~200k in total).

In Figure 8, we draw the open-world performance (AUROC) for the two tasks w.r.t the total memory cost (i.e., the required space to save a model's parameters). We can see that NN takes the most memory

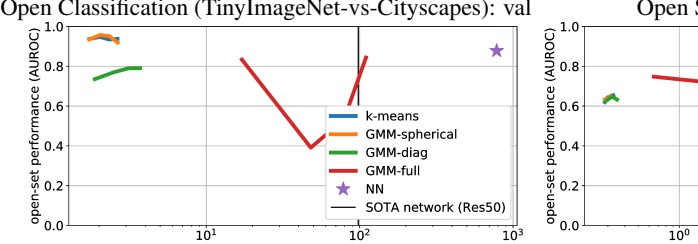

Figure 8: **Open-set performance w.r.t memory cost (MB)** by different models. Memory cost means the space required to store the parameters in these models. **Left**: open-set (200-way) image recognition for TinyImageNet-vs-Cityscapes. **Right**: open-set semantic segmentation (19 classes) on the Cityscapes. NN memorizes training examples for open-world recognition, and hence it consumes huge memory to store OTS features of training examples (more memory consumption than the underlying SOTA models). Compared with the underlying networks, GMM-spherical and k-means induce negligible computation cost. They also perform considerably better than NN and GMM-full/diag on open-set image classification, but not as well as NN and GMM-full on open-world semantic segmentation. These plots clearly serve as guidelines to choose the appropriate statistical models on specific tasks. Note that the validation performance shown here can be nicely translated to test sets, as detailed in Figure 10 .

usage, which is even more than the underlying networks. In contrast, GMM-spherical and k-means models are significantly more compact, i.e., ∼0.3MB for both tasks. Moreover, on open-world image classification (Figure 8-left), k-means and GMM-spherical achieve much better performance than the other models. Interestingly GMM-full achieves the best and stable performance for open-set semantic segmentation (Figure 8-right) but not for open-set image classification (Figure 8-left). Despite this, we note that the validation performance (as plotted here) can be nicely translated to real test sets, as shown in Figure 10).

It is worth noting that the specified PCA-reduced dimension is not optimal that an even lower dimension can lead to better open-world performance (cf. Figure 7). We do not exhaustively explore this in this work, but instead emphasize that our pipeline is quite lightweight that can be tuned on specific tasks, e.g., 0.6MB GMM-full compared with 250MB HRNet for semantic segmentation.

# E   VISUALIZATION OF GAUSSIAN MEANS

As statistical models are interpretable, we visualize what the statistical model can capture. To do so, we visualize per-class Gaussian means through medoid images, which are training images that have features closest to their corresponding per-class mean feature. We show the medoid images in Figure 9, as well as some random images sorted by the cosine similarity (i.e., Euclidean distance on L2-normalized features) to the Gaussian means within each class. We can see the medoid images most likely capture the canonical objects of each class, e.g., those of with "standard" pose and clean background.

# F   OPEN-SOURCE DEMONSTRATION

We attach our code (via three Jupyter Notebook files) to demonstrate our exploration of open-set recognition. One can run the code with access to networks (Res50 and HRNet (Wang et al., 2019)) trained for closed-world tasks. We are not able to upload models or pre-computed features due to space limit, but we are committed to releasing them to the public after paper notification. We refer readers to the Jupyter Notebook files for self-explanatory descriptions.

- "demo_Open-Set-Image-Recognition-Setup-II_GMM_Res50pt_pca_L2norm.ipynb": We show how we train, select and evaluate GMMs on cross-dataset open-set image recognition (Setup-II).
- "demo_tsne_visual_res50pt.ipynb": We show t-SNE visualizations of OTS features of cross-dataset open-set examples (Setup-II). This intuitively demonstrates the benefit of exploiting OTS features for open-set recognition.
- "demo_open-set-semantic-segmentation.ipynb": We demonstrate how we train and evaluate GMM under Setup-III, open-set semantic segmentation.

Table 4: **Cross-dataset evaluation (Setup-II) with a $K$-way classification network that is *trained-from-scratch*.** We report performance with the AUROC metric. This table supplements Table 2. Recall that we train on TinyImageNet as the closed-set, use another dataset as outlier set to tune and select model, and report on the third dataset as the open-set. All the methods operate on off-the-shelf features extracted from the underlying classification network. We report their averaged performance and standard deviation in the last two columns.

| outlier set | MNIST | | | | SVHN | | | | CIFAR | | | | Cityscapes | | | | avg | std |
|---|---|---|---|---|---|---|---|---|---|---|---|---|---|---|---|---|---|---|
| open test set | MN | SV | CF | CS | MN | SV | CF | CS | MN | SV | CF | CS | MN | SV | CF | CS | | |
| *MSP* | .709 | .752 | .694 | .739 | .709 | .752 | .694 | .739 | .709 | .752 | .694 | .739 | .709 | .752 | .694 | .739 | .723 | .023 |
| *Entropy* | .712 | .768 | .703 | .753 | .712 | .768 | .703 | .753 | .712 | .768 | .703 | .753 | .712 | .768 | .703 | .753 | .734 | .027 |
| OpenMax | .145 | .324 | .346 | .538 | .145 | .328 | .348 | .545 | .145 | .328 | .348 | .545 | .139 | .277 | .311 | .789 | .350 | .173 |
| *MSP-calib* | .773 | .803 | .750 | .862 | .773 | .803 | .750 | .862 | .773 | .803 | .750 | .862 | .773 | .803 | .750 | .862 | .797 | .042 |
| *MC-dropout* | .657 | .821 | .725 | .854 | .655 | .841 | .749 | .890 | .659 | .830 | .739 | .872 | .655 | .841 | .749 | .890 | .777 | .086 |
| C2AE | .864 | .744 | .683 | .708 | .864 | .744 | .683 | .708 | .757 | .701 | .755 | .797 | .757 | .701 | .755 | .797 | .752 | .055 |
| GDM | .459 | .859 | .713 | .725 | .444 | .862 | .712 | .724 | .457 | .822 | .712 | .725 | .457 | .822 | .712 | .725 | .683 | .142 |
| *GDM-L2* | .799 | .991 | .886 | .650 | .799 | .991 | .886 | .650 | .799 | .991 | .886 | .650 | .799 | .991 | .886 | .650 | .832 | .125 |
| *NN* | .966 | .993 | .852 | .559 | .966 | .993 | .852 | .559 | .966 | .993 | .852 | .559 | .966 | .993 | .852 | .559 | .843 | .172 |
| *NCM* | .961 | .994 | .963 | .839 | .961 | .994 | .963 | .839 | .961 | .994 | .963 | .839 | .961 | .994 | .963 | .839 | .939 | .059 |
| *kmeans* | .980 | .993 | .923 | .928 | .947 | .998 | .974 | .906 | .979 | .995 | .975 | .833 | .849 | .943 | .877 | .945 | .940 | .050 |
| *GMM* | .987 | .999 | .979 | .905 | .987 | .999 | .979 | .905 | .973 | .999 | .985 | .821 | .811 | .938 | .817 | .928 | .938 | .065 |
| CLS-2 | .999 | .362 | .428 | .514 | .978 | .999 | .985 | .380 | .983 | .988 | .999 | .504 | .959 | .621 | .530 | .999 | .764 | .260 |

Table 5: **Cross-dataset evaluation (Setup-II) with a $K$-way classification network that is *finetuned from an ImageNet pre-trained model*.** We report performance with the AUROC metric. This table supplements Table 2. Recall that we train on TinyImageNet as the closed-set, use another dataset as outlier set to tune and select model, and report on the third dataset as the open-set. All the methods operate on off-the-shelf features extracted from the underlying classification network. We report their averaged performance and standard deviation in the last two columns.

| outlier set | MNIST | | | | SVHN | | | | CIFAR | | | | Cityscapes | | | | avg | std |
|---|---|---|---|---|---|---|---|---|---|---|---|---|---|---|---|---|---|---|
| open test set | MN | SV | CF | CS | MN | SV | CF | CS | MN | SV | CF | CS | MN | SV | CF | CS | | |
| *MSP* | .775 | .770 | .725 | .751 | .775 | .770 | .725 | .751 | .775 | .770 | .725 | .751 | .775 | .770 | .725 | .751 | .755 | .020 |
| *Entropy* | .789 | .787 | .732 | .762 | .789 | .787 | .732 | .762 | .789 | .787 | .732 | .762 | .789 | .787 | .732 | .762 | .768 | .023 |
| OpenMax | .458 | .125 | .473 | .515 | .458 | .125 | .473 | .515 | .458 | .125 | .473 | .515 | .438 | .116 | .464 | .625 | .397 | .164 |
| *MSP-calib* | .832 | .863 | .791 | .851 | .832 | .863 | .791 | .851 | .832 | .863 | .791 | .851 | .832 | .863 | .791 | .851 | .834 | .027 |
| *MC-dropout* | .801 | .783 | .809 | .868 | .801 | .783 | .809 | .868 | .801 | .783 | .809 | .868 | .801 | .783 | .809 | .868 | .815 | .032 |
| C2AE | .842 | .747 | .753 | .762 | .781 | .791 | .766 | .791 | .781 | .791 | .766 | .791 | .781 | .791 | .766 | .791 | .781 | .021 |
| GDM | .766 | .817 | .831 | .633 | .717 | .842 | .845 | .637 | .717 | .842 | .845 | .637 | .690 | .780 | .831 | .696 | .758 | .079 |
| *GDM-L2* | .957 | .999 | .961 | .513 | .957 | .999 | .961 | .513 | .957 | .999 | .961 | .513 | .957 | .999 | .961 | .531 | .857 | .198 |
| *NN* | .901 | .994 | .927 | .715 | .901 | .994 | .927 | .715 | .901 | .994 | .927 | .715 | .901 | .994 | .927 | .715 | .884 | .103 |
| *NCM* | .979 | .995 | .975 | .833 | .979 | .995 | .975 | .833 | .979 | .995 | .975 | .833 | .979 | .995 | .975 | .833 | .946 | .065 |
| *kmeans* | .966 | .998 | .950 | .901 | .966 | .998 | .950 | .901 | .961 | .994 | .963 | .839 | .957 | .981 | .927 | .904 | .948 | .042 |
| *GMM* | .973 | .998 | .953 | .848 | .977 | .999 | .955 | .875 | .957 | .993 | .964 | .837 | .949 | .997 | .967 | .908 | .945 | .050 |
| CLS-2 | .999 | .501 | .500 | .500 | .999 | .999 | .999 | .567 | .999 | .999 | .999 | .522 | .948 | .306 | .519 | .994 | .772 | .256 |

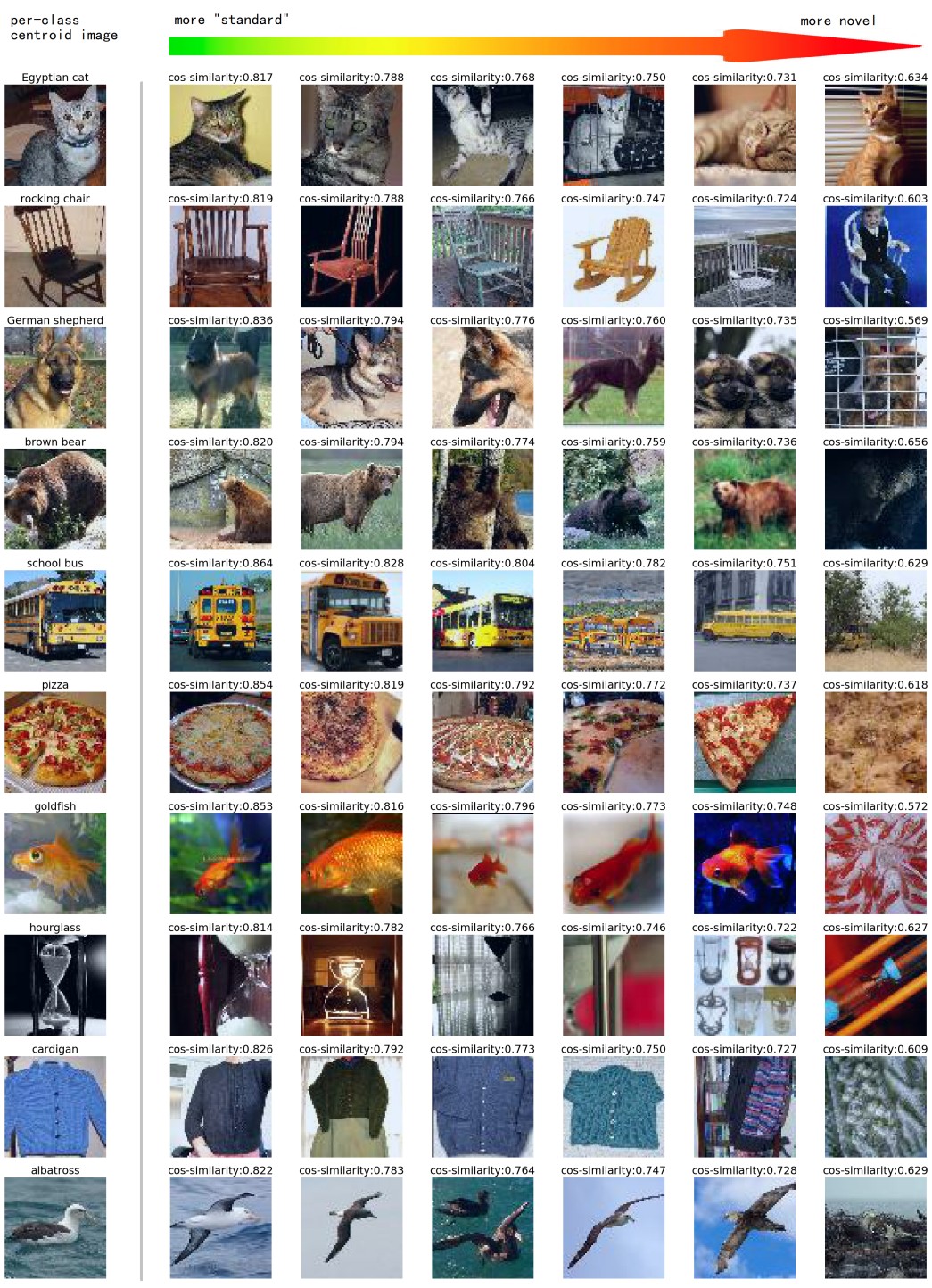

Figure 9: **Visualization of per-class Gaussian means with medoid images** (whose features are closest to the means within their corresponding classes). As comparison, we show some random images sorted by their cosine similarity to the per-class Gaussian mean. We can see the medoid images capture "canonical" objects representing the corresponding classes, e.g., those with "standard" shape and cleaner background. This visualization suggests our statistical models are quite interpretable.

Left table:

| | Res18sc | | Res18pt | | Res50sc | | Res50pt | |
|---|---|---|---|---|---|---|---|---|
| softmax | 0.68709 0.69665 | | 0.77243 0.79142 | | 0.71594 0.73903 | | 0.80152 0.75116 | |
| entropy | 0.69992 0.70584 | | 0.79019 0.81399 | | 0.73473 0.75311 | | 0.82221 0.76174 | |
| openmax | 0.80722 0.82845 | | 0.72989 0.78912 | | 0.69383 0.74160 | | 0.58698 0.60715 | |
| | raw feat. | w/ L2 | raw feat. | w/ L2 | raw feat. | w/ L2 | raw feat. | w/ L2 |
| NN | 0.55936 0.49653 | 0.83039 0.87785 | 0.74432 0.71533 | 0.89929 0.89293 | 0.60031 0.36409 | 0.81536 0.73615 | 0.67506 0.44824 | 0.87807 0.80267 |
| class centroid | 0.70681 0.75428 | 0.87174 0.89203 | 0.49636 0.50535 | 0.90125 0.87412 | 0.62564 0.76653 | 0.87992 0.80921 | 0.56183 0.73232 | 0.86817 0.81324 |
| g-kmeans k=10 | 0.56537 0.63195 | 0.80807 0.85423 | 0.31896 0.33598 | 0.69900 0.73805 | 0.49708 0.67132 | 0.79985 0.76028 | 0.47253 0.64557 | 0.83630 0.88125 |
| g-kmeans k=20 | 0.59580 0.66853 | 0.84097 0.90087 | 0.36686 0.38392 | 0.82044 0.83602 | 0.55660 0.70256 | 0.82518 0.77344 | 0.50895 0.67367 | 0.86058 0.91867 |
| g-kmeans k=30 | 0.61545 0.69318 | 0.82982 0.89229 | 0.34909 0.38631 | 0.77272 0.83650 | 0.58879 0.73276 | 0.85780 0.81856 | 0.55499 0.70158 | 0.91478 0.90814 |
| g-kmeans k=50 | 0.62107 0.68961 | 0.81340 0.87116 | 0.36996 0.42349 | 0.82036 0.87299 | 0.62085 0.76247 | 0.91415 0.89770 | 0.55499 0.73415 | 0.89344 0.91676 |
| c-kmeans k=1 | 0.70186 0.75140 | 0.87360 0.89273 | 0.55047 0.55980 | 0.92296 0.90318 | 0.66943 0.78888 | 0.96877 0.94158 | 0.61114 0.75556 | 0.93691 0.92117 |
| c-kmeans k=3 | 0.73398 0.79438 | 0.84561 0.89350 | 0.55735 0.58189 | 0.92168 0.91595 | 0.69371 0.83714 | 0.91765 0.93243 | 0.66485 0.80758 | 0.94764 0.94443 |
| c-kmeans k=5 | 0.72070 0.78682 | 0.85678 0.89345 | 0.55381 0.58311 | 0.90512 0.90844 | 0.67709 0.83047 | 0.90638 0.93009 | 0.66551 0.80421 | 0.93444 0.93279 |
| c-kmeans k=7 | 0.74315 0.79146 | 0.88913 0.90096 | 0.55758 0.57609 | 0.89558 0.88773 | 0.68635 0.81991 | 0.88040 0.88030 | 0.69303 0.82873 | 0.93786 0.92135 |
| gGMM spherical k=10 | 0.57668 0.62849 | 0.86299 0.89208 | 0.28333 0.30515 | 0.74410 0.79312 | 0.51666 0.67583 | 0.79985 0.76028 | 0.46852 0.65854 | 0.88584 0.85472 |
| gGMM spherical k=20 | 0.64185 0.67571 | 0.87980 0.91619 | 0.30711 0.32998 | 0.90351 0.90201 | 0.55066 0.67288 | 0.83990 0.78481 | 0.52448 0.68474 | 0.91439 0.92795 |
| gGMM spherical k=30 | 0.60383 0.66349 | 0.78068 0.87515 | 0.31939 0.35228 | 0.85606 0.86859 | 0.59350 0.73493 | 0.89787 0.85345 | 0.51726 0.69949 | 0.90947 0.87685 |
| gGMM spherical k=50 | 0.63242 0.69052 | 0.84608 0.90576 | 0.33406 0.37880 | 0.80980 0.85680 | 0.62852 0.76493 | 0.91242 0.88181 | 0.53563 0.72095 | 0.92655 0.90311 |
| gGMM diag k=10 | 0.51207 0.55731 | 0.63581 0.64075 | 0.26688 0.27043 | 0.72490 0.69804 | 0.46097 0.66066 | 0.61491 0.65750 | 0.36070 0.54817 | 0.54244 0.51384 |
| gGMM diag k=20 | 0.54636 0.59195 | 0.65917 0.68460 | 0.29878 0.29888 | 0.78672 0.76459 | 0.47681 0.67541 | 0.65155 0.68746 | 0.37611 0.56186 | 0.56941 0.54173 |
| gGMM diag k=30 | 0.56630 0.60158 | 0.67530 0.69377 | 0.30415 0.31490 | 0.80124 0.77786 | 0.48745 0.68442 | 0.68126 0.71159 | 0.38558 0.57224 | 0.60395 0.57321 |
| gGMM diag k=50 | 0.59710 0.64362 | 0.75863 0.77914 | 0.33509 0.34698 | 0.83754 0.82956 | 0.51583 0.70727 | 0.71937 0.75833 | 0.40744 0.59527 | 0.65055 0.62057 |
| gGMM full k=10 | 0.64417 0.71667 | 0.80195 0.84631 | 0.42858 0.42824 | 0.84978 0.85458 | 0.68250 0.82732 | 0.87916 0.90853 | 0.67047 0.82993 | 0.89235 0.88588 |
| gGMM full k=20 | 0.68907 0.74545 | 0.84687 0.83453 | 0.50331 0.48823 | 0.89736 0.89402 | 0.73841 0.82203 | 0.87474 0.87399 | 0.70719 0.83642 | 0.90174 0.91294 |
| gGMM full k=30 | 0.71424 0.75221 | 0.81392 0.84708 | 0.52625 0.51603 | 0.88250 0.87895 | 0.76597 0.86417 | 0.87427 0.86407 | 0.72177 0.85943 | 0.87070 0.87541 |
| gGMM full k=50 | 0.74406 0.77875 | 0.83715 0.85636 | 0.52806 0.54299 | 0.89174 0.88479 | 0.77377 0.87302 | 0.88350 0.88484 | 0.71656 0.85905 | 0.89419 0.90857 |
| cGMM spherical k=1 | 0.70228 0.75110 | 0.87878 0.89963 | 0.55158 0.55964 | 0.91643 0.90932 | 0.66999 0.78645 | 0.96969 0.93852 | 0.61710 0.75796 | 0.93510 0.92087 |
| cGMM spherical k=3 | 0.72824 0.79507 | 0.87949 0.90907 | 0.55538 0.58032 | 0.92104 0.92129 | 0.69548 0.83311 | 0.94660 0.93824 | 0.66096 0.79863 | 0.95723 0.94063 |
| cGMM spherical k=5 | 0.71568 0.78252 | 0.86419 0.90122 | 0.56363 0.58876 | 0.90670 0.91827 | 0.70371 0.83258 | 0.93394 0.91365 | 0.65688 0.81817 | 0.95136 0.93578 |
| cGMM spherical k=7 | 0.72357 0.77471 | 0.87653 0.89735 | 0.55843 0.58570 | 0.89343 0.90284 | 0.70942 0.84110 | 0.91594 0.89626 | 0.59221 0.75451 | 0.91890 0.93642 |
| cGMM diag k=1 | 0.64393 0.69672 | 0.77731 0.79254 | 0.52407 0.53136 | 0.89267 0.88411 | 0.57466 0.74801 | 0.80679 0.81306 | 0.49017 0.66966 | 0.73538 0.71678 |
| cGMM diag k=3 | 0.67230 0.74967 | 0.81437 0.82493 | 0.52007 0.54376 | 0.89467 0.88792 | 0.60536 0.78359 | 0.81422 0.83336 | 0.52277 0.70216 | 0.76976 0.75729 |
| cGMM diag k=5 | 0.68865 0.74989 | 0.79871 0.81055 | 0.53701 0.55510 | 0.86022 0.85811 | 0.61399 0.79590 | 0.82376 0.83612 | 0.53207 0.72104 | 0.78968 0.77826 |
| cGMM diag k=7 | 0.70357 0.74627 | 0.81697 0.81874 | 0.54118 0.54688 | 0.85723 0.85988 | 0.62480 0.79993 | 0.80843 0.81800 | 0.53032 0.71588 | 0.79067 0.80145 |
| cGMM full k=1 | 0.72876 0.78914 | 0.81531 0.84522 | 0.58701 0.58414 | 0.87386 0.87107 | 0.73569 0.88405 | 0.86319 0.88178 | 0.67033 0.82037 | 0.83378 0.83184 |
| cGMM full k=3 | 0.70588 0.76389 | 0.81569 0.83442 | 0.55963 0.57559 | 0.84525 0.85142 | 0.35908 0.50734 | 0.27616 0.37136 | 0.37132 0.50888 | 0.39130 0.45191 |
| cGMM full k=5 | 0.67757 0.77610 | 0.81195 0.85312 | 0.50838 0.53217 | 0.81207 0.84090 | 0.45285 0.55336 | 0.65844 0.65660 | 0.30321 0.39585 | 0.53254 0.57065 |
| cGMM full k=7 | 0.67518 0.73784 | 0.79417 0.84090 | 0.44711 0.48778 | 0.80688 0.81082 | 0.49434 0.60215 | 0.79676 0.75176 | 0.49732 0.57812 | 0.84393 0.85776 |

Right table:

| | raw feat. | w/ L2 |
|---|---|---|
| softmax | 0.60226 0.59017 | |
| entropy | 0.57166 0.59956 | |
| openmax | 0.61358 0.65467 | |
| NN | 0.50528 0.49209 | 0.67170 0.76910 |
| class centroid | 0.50273 0.48175 | 0.61522 0.71535 |
| kmeans k=1 | 0.55085 | 0.62010 |
| kmeans k=3 | 0.60075 | 0.64429 |
| kmeans k=5 | 0.60218 | 0.64907 |
| kmeans k=7 | 0.61918 | 0.65465 0.75522 |
| GMM spherical k=1 | 0.53039 | 0.63315 |
| GMM spherical k=3 | 0.62693 | 0.63907 |
| GMM spherical k=5 | 0.62961 | 0.65128 |
| GMM diag k=1 | 0.53820 | 0.62669 |
| GMM diag k=3 | 0.61556 | 0.64717 |
| GMM diag k=5 | 0.62053 | 0.63280 |
| GMM full k=1 | 0.70429 | 0.74774 0.79511 |
| GMM full k=3 | 0.71349 | 0.72420 |
| GMM full k=5 | 0.71389 | 0.73348 0.77999 |

Figure 10: Detailed results of (left) open-set image recognition using Cityscapes images as cross-dataset open-world examples under Setup-II) and (right) open-set semantic segmentation (Setup-III). In each cell, the first and second (if existing) row numbers denote AUROC performance on the val and test sets, respectively. We highlight the best performance on the val set, on which we tune the hyper-parameter and report the performance on the test set. As for notation, gGMM means we learn GMM "globally" on the whole closed train-set, agnostic to class labels; while cGMM means that we learn class-conditional GMMs. For open-set semantic segmentation, we only train GMM in a class-conditional fashion (i.e., cGMM), because using pixel features from all classes to train a global GMM is prohibitively time-consuming. "Raw feat." means the feature we extract from the last convolution layer without L2-normalization, and "w/ L2" means we L2-normalize the extracted features. Clearly, L2-normalization greatly boosts open-world performance.

