# OpenReview forum: "An Empirical Exploration of Open-Set Recognition via Lightweight Statistical Pipelines"
_ICLR.cc/2021/Conference — Reject_

### Official Review · AnonReviewer4 · 2020-10-27
**Interesting and intuitive approach for open-set recognition**

**Rating:** 7
**Confidence:** 4

**Review:**

This paper proposes an open-set recognition approach that uses simple statistical measures (such as GMM’s and KMeans) on top of post-processed intermediate features extracted from closed-set deep models. It finds that i) this “lightweight” pipeline outperforms prior methods on open-set image recognition across multiple evaluation protocols at much lesser memory and compute cost, and ii) Open-world recognition generally benefits from using models pre-trained on large datasets such as ImageNet rather than training from scratch, iii) the technique also generalizes to open-world semantic segmentation.

Strengths

– The paper studies an important problem, is motivated clearly, and is well-written

– The proposed approach is intuitive, memory-efficient, and appears to clearly and consistently perform prior work

– The modeling choices are motivated and analyzed well, for eg. Fig 3 clearly illustrates why l2-normalized and pooled intermediate features more clearly capture in-domain-ness than logits used predominantly in prior work

– The set of experiments and baseline comparisons appears comprehensive

– The performance v/s memory tradeoff of the current method vs prior work is analyzed well

– The limitations of the proposed method are also explored, for eg. Fig 5 that shows how for open set semantic segmentation, the proposed method is a better alternative than training a binary classifier only when very little out-of-domain data is available to train on

Weaknesses

– Studying how/whether the choice of layer from which features are extracted affects performance would have been an interesting addition

Additional comments / suggestions

– Fig 6 is difficult to read since some of the lines overlap, varying opacity might help with readability

Overall comments

This is an interesting and well-written paper that proposes simple and memory-efficient alternatives for open-world recognition that consistently outperform more complex methods from prior work.

-------- post-rebuttal comments --------

I have read the concerns raised by other reviewers as well as the author response. I still feel that this is an interesting work and its findings are of potential value to the community: There has been a considerable amount of recent work in open set recognition for deep models, and this paper calls into question the need for sophisticated techniques by showing (fairly rigorously, in my opinion) that simple strategies and the right choice of feature engineering works better. I agree with the authors that not using ImageNet pretraining is an unrealistic and unnecessary constraint – moreover, the method generalizes even when evaluated on datasets such as MNIST and SVHN, which are distributionally very different from ImageNet. Further, I think the paper acknowledges prior work appropriately and did not find any of the claims made to be unreasonable. I agree with R1's concerns about overlap with two recent papers, but found the author response to be satisfactory. Overall, I will retain my accept rating.

---

> ### Author Response · Authors · 2020-11-25
> **Response**
>
> Thank you for the positive comments!
>
> **1. "Studying how/whether the choice of layer from which features are extracted affects performance would have been an interesting addition".**
>
> We agree! We suspect that lower layer features may reveal outlier images with different appearance statistics - eg, a perception system for autonomous driving may need to recognize certain rare weather conditions (a hail storm) as anomalous.

---

### Official Review · AnonReviewer3 · 2020-10-29
**A reinventing-the-wheel paper**

**Rating:** 3
**Confidence:** 5

**Review:**

Methodology:
The paper tackles the so-called open-set classification where query examples outside any of the classes in the training set should be detected at inference. By combining the feature extractor based on fashionable deep models and the classical clustering methods (k-means, GMM, etc), the paper empirically shows that this pipeline can address many open-set problems in realistic scenarios.

Pros:
Unfortunately, none.

Cons:
The paper basically reinvents the wheel. The so-called open-set classification problem is precisely the anomaly detection problem that has been studied for decades (see V. Chandola, A. Banerjee, and V. Kumar, “Anomaly detection: A survey,” ACM Computing Surveys, vol. 41, no. 3, p. 15, 2009), and the use of classical clustering methods (k-means, GMM, etc) was among one of the first efforts for anomaly detection. Applying the identical strategy on top of the fashionable deep features is the routine treatment for anomaly detection 101 in 2020, and brings in no novelty to the community.

---

> ### Author Response · Authors · 2020-11-25
> **Response**
>
> Thank you for the recommended survey paper on anomaly detection! We have cited in our updated version.
>
> **1. The so-called open-set classification problem is precisely the anomaly detection problem that has been studied for decades.**
>
> Our related work section readily acknowledges many related problems in the general space of open-world recognition such as out-of-distribution detection, anomaly detection, and open-set classification. Open-set classification differs from the others in that it also evaluates closed-world classification performance on the classes of interest. In our case, we build our solutions on features extracted from closed-world networks that are already state-of-the-art, ensuring that our closed-world performance also remain so. Note that we do not introduce the problem of open-set classification, but rather build on an existing body of work.
>
> **2. The paper has no novelty as GMM has already been studied in anomaly detection (Chandola et al, 2009).**
>
> By no means do we claim that GMMs are novel. As stated in our title, we focus on an empirical exploration of "insufficiently well-known best practices" (page-2 in the paper) for open-set recognition. We believe that empirical studies that point out key implementation details that allow simple methods to be competitive, can have lasting impact - cf (Hendrycks and Gimpel, ICLR 2017) and (Hendrycks et al. ICLR 2019). At the very least, we believe our work suggests lightweight baselines moving forward (that have been ignored in contemporary benchmark studies).  Finally, because our solutions are admittedly quite simple and lightweight, they also may be easier to deploy for low latency, safety critical ML tasks (such as autonomous driving).

---

### Official Review · AnonReviewer1 · 2020-10-30
**Paper empirically explores open set classification. It claims that leveraging pre-training information and classical learning schemes with appropriately chosen feature perform better than OOD schemes.  The experimental approach and results are not convincing.**

**Rating:** 3
**Confidence:** 3

**Review:**

Summary of Paper: The main claim of the paper is that out of distribution (OOD) detection can be done by use of pre-training and appropriately deriving a feature space from SOTA activations  via pooling, PCA based dimensionality reduction, L2 normalization.  Classical methods such as GMMs, k-means etc. can then be used to estimate the probability density function of features for use in OOD detection.  Several alternative schemes are compared against many OOD detection schemes.

Key contributions claimed are that pre-trained nets have information about open-world statistics and off-the-shelf net features along with appropriate choice of a low-dimensional representation helps in outperforming conventional OOD schemes.
The paper builds upon recent work on OOD method benchmarking method by Shafaei et al (2019) that argues that most OOD schemes are not able to pass a less unbiased test designed ( wherein a Source data set is used for training using standard methodology, Validation data set for estimating a decision function between source and validation,  and finally the probability of outlier detection on other datasets and their variability is estimated to get robust view of OOD.).

1. The general approach is, as I think, quite misguided: the idea is to use pretrained datasets, extract features from it and apply a Gaussian mixture model. This means though, that such an approach will be able to recognize only classes for which features were made available (hence, the closed-world problem is just extended by additional knowledge about more classes from selected datasets, no open-world problem is solved).

2. The fact that pretraining helps, is a generic statement which depends on the statistics of the dataset used for pretraining and then the dataset on which it is tested.

3. the setup in 4.1 (split a single dataset into open and closed sets w.r.t class label) is questionable - as the image statistics in the partition are same.

4. in 4.2 the authors use an 'open' dataset for validation/tuning - this makes this dataset not open per definition. It may be true that it helps in the model generalizing better, but the terminology is still misguided.  In my view, the paper applies the testing methodology of Shafaei et al (2019) incorrectly to design an OOD algorithm and claim its superiority.

None of the tables in the paper provide error bars.  In order to convince that the insights are correct I would expect that the experiments need to be run with a larger sampling of outlier datasets (as done in Shafaei et al).

---

> ### Author Response · Authors · 2020-11-25
> **Response**
>
> Thank you for your comments!
>
> **1. Pretraining just extends the closed-world problem by additional knowledge about new classes, but does not solve open-world problems.**
>
> We agree that pretraining may not naturally fit into current open-world protocols (that require certain objects to be never-before-seen). But we humbly argue that such protocols may be too artificial for real-world deployment. Consider a perception system for a self-driving car. If pre-training on diverse, *non-urban* datasets produces robust ML systems that safely operate in urban scenes, we argue this is a good thing!
>
> **2. "Pretraining helps" is a generic statement which depends on the statistics of the dataset used for pretraining and then the dataset on which it is tested.**
>
> We completely agree! This statement is indeed an empirical finding that depends on the data at hand. That's why we empirically validate it, both in Setup-III (open world segmentation) and cross-datasets in Setup-II (Table 2). For example, we use Imagenet for pre-training, SVHN as an outlier dataset for valiation, and MNIST as the held-out open-set for testing. This is a less biased setup in which the pre-training ImageNet will not likely be similar to any of open-set datasets (MNIST, SVHN and Cityscapes).
>
> **3. Setup-I in Section 4.1 is questionable that splits a single dataset into open and closed sets w.r.t class labels. The image statistics in the partition is the same.**
>
> We completely agree! That's why we explore other setups, II and III. As stated in Section 4.1, we evaluate Setup-I only because it is a well-established protocol used by prior work in open-set recognition, facilitating comparisons to a large body of methods.
>
> **4. "In my view, the paper applies the testing methodology of Shafaei et al (2019) incorrectly to design an OOD algorithm and claim its superiority".**
>
> We think Reviewer1 mistakenly believes that we use the same dataset distribution for validation and test, but we sample from different datasets, as stated in Section 4.2 of the paper (as well as our code in the supplementary material, cf. Appendix F). We think our setup directly follows Shafaei et al, who train from one outlier distribution (A) and tests on another (B).  In our case, we validate instead of directly training on the outliers from (A).
>
> **5. Using the terminology "open training data" is misguided.**
>
> We agree that the terminology in this space can be confusing. We now use "outlier training data" in the updated paper to be consistent with Shafaei et al (2019), but appreciate any suggestions.
>
> **6. Tables do not have error bars.**
>
> We have added error bars in Table 1, and provided two extra tables (Table 4 and 5) in the appendix that have error bars.

---

### Official Review · AnonReviewer2 · 2020-11-03
**Good Empirical Study Falls Short**

**Rating:** 4
**Confidence:** 4

**Review:**

## Summary and Contributions
The paper presents an empirical study of the open set image recognition (OSR) problem. It considers two classification setups -- single-domain and cross-domain, and, one semantic segmentation setup. In particular, the paper evaluates simple statistical models like the Nearest Class Means (NCMs), KMeans, and, Gaussian Mixture Models (GMMs) built atop OTS deep features against deep SOTA baselines like CLS^(K+1), MSP, OpenMax, C2AE, GDM, etc.

The paper demonstrates that simple statistical models outperform SOTA models if deep OTS features pretrained on large datasets are used, normalized and their dimensionality is reduced through spatial pooling and PCA. This also results in really light-weight image recognition pipelines for OSR settings.

## Detailed Review
The following is the detailed review of the paper, organized into strengths and weaknesses subsections.

### Strengths
#### Relevance and Significance
Most ML models, when deployed in real-world settings, often need to operate in open-set (or, open world) conditions and need to provide robust estimates including an 'out-of-class'/ 'unknown' label when encountering data from unknown classes. Models that have good OSR properties should be of broad interest to the ML community.

Empirical results in the paper suggest that simple statistical techniques can be quite effective in performing open-set recognition. The paper also validates the significance of performing feature preprocessing steps towards obtaining performance on par with (or exceeding) that of more complex SOTA methods.

#### Clarity
The paper is written well and is easy to understand.

### Weaknesses
#### Relation to Prior Art
The paper does a reasonable job of presenting the prior art, identifying the challenges and need for the presented work. However, it doesn't cite the following relevant works:

[1] Sun et al, Conditional Gaussian Distribution Learning for Open Set Recognition, CVPR 2020.

[2] Geng et al, Recent Advances in Open Set Recognition: A Survey, IEEE PAMI 2020.

#### Novelty
The study involving the exploration of preprocessing steps on a variety of simple statistical models seems novel. However, the survey paper by Geng et al [2] also conducts an empirical study, involving many of the models considered in this paper. The paper needs to reference and compare against [2].

#### Empirical Evaluation
The empirical evaluation is inadequate. Experiment results are shown only for limited closed sets for training. Being an empirical paper it is expected for the paper to have more detailed experiments. In addition,

(1) The paper needs to redesign the study in light of [2]

(2) It should compare results against more recent SOTA models, for example, [1] given that both uses class-conditional GMMs.

(3) Authors mention in Section 4 that hyperparameters are selected based on a small-scale validation set. However, the size of this validation set is not specified. Details about the validation set are needed to determine performance of the experiments.

(4) In Section 4.1, the C2AE scores in Table 1 seems to be lower than what is reported in their paper, however experimental setup the same: https://arxiv.org/pdf/1904.01198.pdf. Is there any specific reason for this?

(5) Section 4.2: Results on more than one close-world training dataset including MNIST, CIFAR for cross-world testing should be provided as also done by Shafaei et al.

(6) The experimental results need to be explained better. In sections 4.1 and 4.3, GMMs gives good results while on the setting of section 4.2, GMMs don't do well. What is the explanation for this?

## Assessment
This paper presents a study showing that simple, lightweight, statistical models can outperform deep SOTA models on the image OSR problem. This should be of interest to the ML community. However, the paper seems agnostic of a couple of recent works which render the study and the empirical evaluation inadequate. In addition, experimental evaluation and analysis needs to be improved as per the observations made above.

---

> ### Author Response · Authors · 2020-11-25
> **Response**
>
> Thank you for the recommended papers! We have cited them in our updated paper.
>
> **1. The paper should compare to more methods covered by the survey paper (Geng et al 2020).**
>
> Table 1 in our updated draft now includes more methods from Table 5 in (Geng et al 2020).
>
> **2. The paper should compare to more recent SOTA methods like (Sun et al 2020).**
>
> We have cited and compared against (Sun et al 2020), as well as more methods (cf. the last paragraph in page-5). While these methods do outperform ours, we stress that our goal is an empirical study of crucial details in open-set recognition (pretraining, normalization, feature pooling) that make simple statistical methods (e.g., GMM) surprisingly competitive.
>
> **3. The paper needs to redesign the study in light of (Geng et al 2020).**
>
> Our Setup-I is exactly the experimental design in (Geng et al 2020) which splits a single dataset (e.g., YaleB and COIL20) into open- and closed-sets w.r.t class labels. As Reviewer1 states, such an empirical protocol may arguably be too artificial. This is why we focus on Setup-II (cross-dataset evaluation) and Setup-III (a larger-scale study through semantic segmentation), neither of which are explored in (Geng et al 2020). Therefore, we believe our experimental design to be more complete and inclusive.
>
> **4. What is the size of the small-scale validation set.**
>
> Specifically, for Setup-II, we use 10 images as the validation set (as stated in Section 4.3). Comparing to 2965 and 500 images in the training and testing sets, the validation set is small. In the updated paper, we have removed the term "small-scale" to avoid confusion.
>
> **5. C2AE scores are lower than what reported in the main paper.**
>
> We acknowledge this concern, as we devoted considerable effort to addressing the disconnect between our reproduced numbers and the reported C2AE numbers. As there is no publicly available code for C2AE, we previously reimplemented C2AE and saw a worse performance (e.g., 0.78 AUROC on CIFAR10). Because of this, we reached out to the authors who validate our reimplementation (hence the footnote-1). In the updated Table 1, we directly copy their reported number.
>
> **6. In terms of Setup-II, the paper should include more than one training sets (e.g., MNIST and CIFAR10) as done by Shafaei et al (2019).**
>
> We choose TinyImageNet because it has the largest scale (100k) and largest number of classes (200) compared to other datasets (MNIST and CIFAR10). Moreover, from (Shafaei et al 2019), methods degrade more notably on TinyImageNet than other datasets (including MNIST and CIFAR10). Therefore, we believe conclusions drawn from training with TinyImagenet are reliable.
>
> **7. GMMs don't do well in Section 4.2.**
>
> We note that GMMs (0.945 AUROC) do perform well compared to the best-performing method kmeans (0.948 AUROC), which can be seen as a special case of GMMs that have a (infinitesimally) small-scale identity covariance. We conjecture that learning the covariance in GMMs can sometimes overfit. In practice, one can perform model selection (kmeans or GMMs with full-rank covariance, diagonal covariance, etc) on the val set, as studied in Figure 10. We have added such an analysis in the updated paper (Section 4.2, page 7).

---

### Decision · Program_Chairs · 2021-01-07
**Final Decision**

**Decision:**

Reject

**Comment:**

The addresses open-set recognition, namely, detecting anomalous samples that belong to classes not observed during training.
It has been shown that existing methods fail on open-world images. The current paper shows empirically that performance can be greatly improved if based on low-dimensional features.

Reviewers had grave concerns about the novelty of the approach and the logic behind the workflow. They found merit in the paper but chose to retain their scores after reviewing the rebuttal. As a future recommendation, it would be useful to provide more evidence about what component of the method or workflow are novel and what makes them work well.